# Loss of Amphiregulin drives inflammation and endothelial apoptosis in pulmonary hypertension

Jonathan Florentin[1], Jingsi Zhao[1], Yi-Yin Tai[1], Wei Sun[1], Lee L Ohayon[1], Scott P O'Neil[1], Anagha Arunkumar[1], Xinyi Zhang[1], Jianhui Zhu[3], Yassmin Al Aaraj[1], Annie Watson[1], John Sembrat[1,2], Mauricio Rojas[1,2], Stephen Y Chan[1,*], Partha Dutta[1,4,*]

Pulmonary hypertension (PH) is a vascular disease characterized by elevated pulmonary arterial pressure, leading to right ventricular failure and death. Pathogenic features of PH include endothelial apoptosis and vascular inflammation, which drive vascular remodeling and increased pulmonary arterial pressure. Re-analysis of the whole transcriptome sequencing comparing human pulmonary arterial endothelial cells (PAECs) isolated from PH and control patients identified *AREG*, which encodes Amphiregulin, as a key endothelial survival factor. PAECs from PH patients and mice exhibited down-regulation of *AREG* and its receptor epidermal growth factor receptor (EGFR). Moreover, the deficiency of *AREG* and *EGFR* in ECs in vivo and in vitro heightened inflammatory leukocyte recruitment, cytokine production, and endothelial apoptosis, as well as diminished angiogenesis. Correspondingly, hypoxic mice lacking *Egfr* in ECs (*cdh5*[cre/+] *Egfr*[fl/fl]) displayed elevated RVSP and pulmonary remodeling. Computational analysis identified *NCOA6*, *PHB2*, and *RRP1B* as putative genes regulating *AREG* in endothelial cells. The master transcription factor of hypoxia HIF-1$\alpha$ binds to the promoter regions of these genes and up-regulates their expression in hypoxia. Silencing of these genes in cultured PAECs decreased inflammation and apoptosis, and increased angiogenesis in hypoxic conditions. Our pathway analysis and gene silencing experiments revealed that BCL2-associated agonist of cell death (BAD) is a downstream mediator of *AREG*. *BAD* silencing in ECs lacking *AREG* mitigated inflammation and apoptosis, and suppressed tube formation. In conclusion, loss of Amphiregulin and its receptor EGFR in PH is a crucial step in the pathogenesis of PH, promoting pulmonary endothelial cell death, influx of inflammatory myeloid cells, and vascular remodeling.

## Introduction

Inflammation is a biological process mediated by immune cells such as neutrophils, monocytes and macrophages, and soluble molecular mediators like IL-6, IFN-$\beta$, and TNF-$\alpha$ (Iqbal et al, 2016; Vasamsetti et al, 2018). Inflammation is a defensive response against various stimuli such as pathogens and irritants. However, inflammation often propagates pathogenesis in various inflammatory diseases such as diabetes, atherosclerosis, and pulmonary hypertension (PH) (Tuder et al, 1994; Galkina & Ley, 2009; Price et al, 2012; Graham et al, 2013, 2018; Pugliese et al, 2015, 2017; Tsalamandris et al, 2019; Vasamsetti et al, 2020). PH is a progressive cardiopulmonary vascular disease, characterized by elevated mean pulmonary arterial pressure at rest and lung vascular remodeling. PH and its severe subtype pulmonary arterial hypertension (PAH) are thought to be driven by excessive expansion of smooth muscle cells, apoptosis of pulmonary artery endothelial cells (PAECs) (Tuder, 2017), and exaggerated inflammation (Rabinovitch et al, 2014; Amsellem et al, 2017; Florentin et al, 2018, 2021). However, the molecular interconnections linking these cellular phenotypes are not fully understood.

Endothelial dysfunction is crucial to the development of PH (Ranchoux et al, 2018). The pathogenic rise in pulmonary pressure is due to progressive loss of small pulmonary arterioles. The initial triggers caused by genetic or environmental factors can lead to endothelial injury and impaired vascular regeneration. It is thought that during the initial phase of PH, pulmonary vascular ECs are more likely to undergo apoptosis and possibly senescence (Culley et al, 2021) followed by the selection of ECs that exhibit resistance to apoptosis and a hyperproliferative response (Sakao et al, 2005; Michelakis, 2006). Furthermore, endothelial metabolic changes that may regulate EC survival play crucial roles in PH pathogenesis (Bertero et al, 2016). Yet, a knowledge gap exists, regarding the molecular pathways that regulate survival and regeneration of pulmonary ECs, leading to exaggerated inflammation, after vascular injury.

[1]Division of Cardiology, Department of Medicine, Center for Pulmonary Vascular Biology and Medicine, Pittsburgh Heart, Lung, Blood, and Vascular Medicine Institute, University of Pittsburgh School of Medicine, University of Pittsburgh Medical Center, Pittsburgh, PA, USA  [2]Division of Pulmonary, Allergy and Critical Care Medicine, University of Pittsburgh, Pittsburgh, PA, USA  [3]Department of Pathology, University of Pittsburgh, Pittsburgh, PA, USA  [4]Department of Immunology, University of Pittsburgh School of Medicine, Pittsburgh, PA, USA

Correspondence: duttapa@pitt.edu; chansy@pitt.edu
*Stephen Y Chan and Partha Dutta contributed equally to this work.

Our analysis of previously published unbiased whole genome transcriptome data (Rhodes et al, 2015) to discover cell survival genes revealed that PAECs isolated from the lungs of patients with PAH displayed a sixfold reduction in the expression of *AREG* compared with PAECs isolated from age-matched healthy patients. *AREG* encodes Amphiregulin, which mediates cell survival and proliferation (Piepkorn et al, 1994; Enomoto et al, 2009). Prior studies demonstrated the participation of Amphiregulin in a wide variety of physiological processes (Schneider & Wolf, 2009) such as mammary gland development and ductal morphogenesis during puberty. Estrogens induce Amphiregulin, and through the estrogen receptor alpha (ERα), induce proliferation of the mammary epithelium (Ciarloni et al, 2007; LaMarca & Rosen, 2007). Other studies reported the role of Amphiregulin in cancer cell proliferation (Berasain & Avila, 2014). The contributions of this growth factor in inflammatory disease pathogenesis such as psoriasis and rheumatoid arthritis have also been reported (Bhagavathula et al, 2005; Yamane et al, 2008). In addition, Amphiregulin has been shown to increase cardiac fibrosis and aggravate cardiac dysfunction in a mouse model of myocardial infarction (Liu et al, 2018), and promote fibroblast activation in pulmonary fibrosis (Ding et al, 2016; Liu et al, 2016b). The anti-apoptotic and protective roles of epidermal growth factor receptor (EGFR) in diseases, such as acute lung injury and liver injury (Meng et al, 2020; Wu et al, 2020b), have been shown. However, whether the loss of Amphiregulin promotes endothelial apoptosis and vascular inflammation in PH is not known.

In the present study, we aimed to define the function of Amphiregulin and its receptor the EGFR in (A) PAEC survival and proliferation as well as (B) inflammation in PH. To do so, we used computational genomic and multicolor flow cytometric analyses of samples obtained from PAH patients and multiple PAH animal models. We observed significant down-regulation of Amphiregulin and *EGFR* in PAECs in patients with PAH. In vitro, *EGFR* and *AREG* silencing in PAECs increased inflammation and apoptosis and suppressed their proliferation and tube formation ability. Conversely, Amphiregulin exposure dampened inflammation, protected these cells against apoptosis, and encouraged angiogenesis ability. These mechanisms correlated with the findings in vivo of mice lacking *Egfr* in endothelial cells. Pathway analysis identified *NCOA6*, *PHB2*, and *RRP1B* as *AREG* inhibitor genes in endothelial cells. These genes, which were up-regulated in hypoxia, were found to have binding motifs for HIF-1α in their promoter regions. Silencing of these genes in PAECs diminished inflammatory cytokine production and apoptosis as well as increased tube formation ability. In addition, we found that BCL2-Associated Agonist of Cell Death (BAD), a downstream mediator of *AREG*, when silenced, reversed increased inflammation, endothelial apoptosis, and suppressed tube formation observed in absence of *AREG*. Altogether, these results define Amphiregulin and EGFR as PAEC survival factors and potential therapeutic targets to reduce inflammation in PH.

# Results

## *AREG* expression is decreased in PH

It is suggested that PH is characterized by initial PAEC apoptosis followed by hyperproliferation of apoptosis-resistant ECs (Tuder et al, 1994, 2001; Hirose et al, 2000; Nicolls et al, 2005). To understand the mechanisms of PAEC loss in PH, we analyzed RNA sequencing data (dbGaP genotype files: phs000998.v1.p1 NHLBI/iPSC_PulmonaryHypertension) (Chavakis & Dimmeler, 2002; Rhodes et al, 2015) (Fig 1A). The analysis of the RNA-Seq data comparing PAEC of seven control subjects and six patients with idiopathic pulmonary arterial hypertension revealed down-regulation of most genes encouraging EC survival (Fig 1B). Among these pro-survival genes, PAEC isolated from the lungs of patients with PAH had sixfold reduction in the expression of *AREG* compared with those isolated from age-matched healthy patients (Fig 1B). To understand if *AREG* signaling is altered in PH, we cultured human PAECs under hypoxic conditions, and analyzed PAECs in patients with PAH and mice with PH. We identified genes downstream to *AREG* using Ingenuity Pathway Analysis (IPA) (Fig S1A). Correspondingly, hypoxia significantly decreased *AREG* expression as well as these downstream genes in cultured PAECs (Fig 1C). In addition, using immunofluorescence imaging, we showed that AREG expression in pulmonary ECs of hypoxic mice and PAH patients is down-regulated compared with the controls (Fig S1B and C). Amphiregulin encoded by *AREG* binds to the EGFR. Using confocal imaging, we observed down-regulation of EGFR in PAECs of PAH patients (Fig 1D), C57BL/6 mice under hypoxic conditions (Fig 1E), *Il6*[tg] mice (Florentin et al, 2021) (Fig S1D) and monocrotaline-injected rats (Fig S1E) compared with age-matched control patients and mice under normoxic conditions, respectively. Finally, we demonstrated a concomitant decrease in Egfr expression and increase in caspase 3 levels in hypoxic and *Il6*[tg] mice, and in monocrotaline-injected rats (Fig S2A upper panel). Moreover, there were inverse correlations between Egfr and caspase 3 expression (Fig S2A lower panel). In addition, we found an increased Bad expression in pulmonary ECs across all rodent PH models, suggesting that Egfr regulates the expression of Bad, protecting ECs against apoptosis (Fig S2B).

## Deficiency of *EGFR* in ECs aggravates PH

The role of Amphiregulin/EGFR in cardiovascular diseases has been reported (Makki et al, 2013; Peng et al, 2016; Liu et al, 2016a). However, the importance of these molecules in PH remains understudied. Because we observed that *AREG* and *EGFR* are down-regulated in pulmonary endothelial cells in patients and mice with PH, we hypothesized that EC-specific *Egfr* deficiency worsens PH pathogenesis. To this end, we generated mice that lack *Egfr* in EC (*Cdh5*[cre/+] *Egfr*[fl/fl]), notably using *Cdh5*[cre] mice (Payne et al, 2018) that do not incur off-target ablation of expression in immune cells seen with other endothelial-specific Cre mouse lines (De Palma et al, 2005; Chen et al, 2016). These mice exhibited elevated right ventricular systolic pressure (RVSP) (Figs 1F and S2C) and exacerbated lung remodeling (Fig 1G) compared with control *Cdh5*[+/+] *Egfr*[fl/fl] mice under hypoxic conditions. Although normoxic *Cdh5*[cre/+] *Egfr*[fl/fl] had similar RVSP, they displayed worsened lung remodeling compared with *Cdh5*[+/+] *Egfr*[fl/fl] mice housed under normoxic conditions.

## The loss of *AREG* or its receptor *EGFR* increases EC apoptosis

Various studies showed that Amphiregulin and Amphiregulin receptor EGFR play an important role in survival and proliferation of

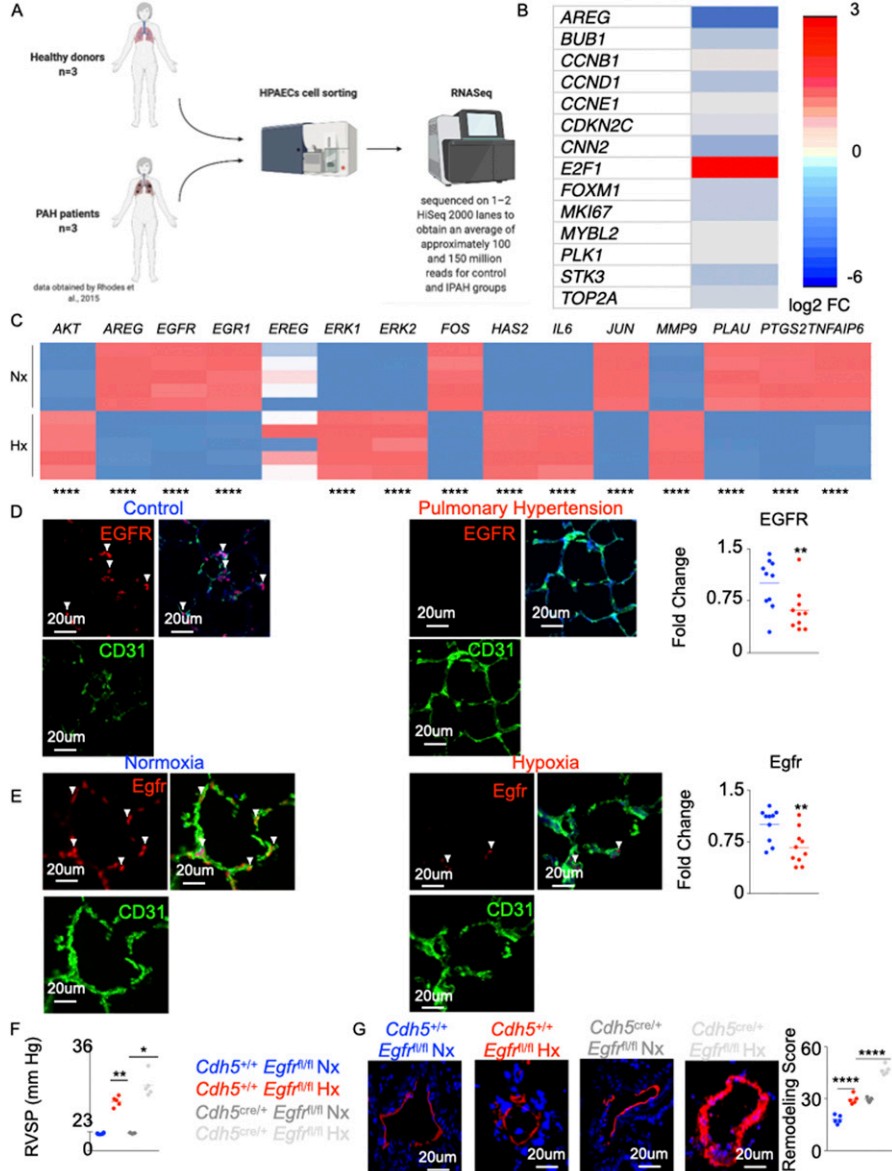

**Figure 1. Deficiency in *AREG/Egfr* aggravates pulmonary hypertension.**
RNA sequencing data (dbGaP genotype files: phs000998.v1.p1 NHLBI/iPSC_PulmonaryHypertension) (Rhodes et al, 2015) were analyzed. Lungs of pulmonary arterial hypertension patients and healthy controls were collected (n = 7 for controls and 6 for pulmonary arterial hypertension patients). Lungs of hypoxic and normoxic mice were harvested (n = 5 for each group). **(A)** The flowchart depicts the analysis of the RNASeq data obtained by Rhodes et al (2015). **(B)** Heat map depicting expression of the cell survival genes (expressed as log$_2$ FC) obtained from the RNASeq analysis. **(C)** Heat map showing expression of the genes downstream to *AREG* using qRT-PCR in human pulmonary endothelial cell line. **(D, E)** Confocal imaging of human (D) and mouse (E) lungs showing expression of EGFR (green) in CD31 (white)-expressing endothelial cells. Smooth muscle actin (SMA) (red) was used to stain the medial layer. The arrows depict EGFR+ vascular endothelial cells. **(F, G)** In normoxic and hypoxic *Cdh5*cre/+ *Egfr*fl/fl and littermate control mice, right ventricular systolic pressure (F) and lung vasculature remodeling (G) were quantified. n = 5 per group. Data are shown as mean. *P < 0.05, **P < 0.01, ****P < 0.001.

vascular smooth muscle cells (Kato et al, 2003) and endothelial cells (Bordoli et al, 2011; Lee et al, 2016), and increase athero-genesis (Dreux et al, 2006; Bordoli et al, 2011). However, the contributions of Amphiregulin and EGFR in the setting of PH are not known. To this end, the expression of *AREG* in PAECs was knocked down using siRNA. Hypoxia-induced endothelial apoptosis was further aggravated in the absence of *AREG* (Fig 2A). Similarly, *AREG* silencing decreased angiogenic tube formation ability (Fig 2B). In line with this observation, mice lacking *Egfr* in ECs (*cdh5*cre/+ *Egfr*fl/fl) displayed decreased number of PAECs under hypoxic conditions (Fig 2C and D). PAECs in these mice also displayed greater apoptotic activity compared with the littermate control mice (*cdh5*+/+ *Egfr*fl/fl) (Fig 2E and F). Concomitantly, we found that ECs from hypoxic *Cdh5*cre/+ *Egfr*fl/fl mice had higher

expression of Ki-67 than the WT controls suggesting that *Egfr*-deficient ECs are hyperproliferative (Fig S2D). Next, we sought to determine the impact of EGFr on endothelial biology in vitro. We silenced *Egfr* in HPAECs in vitro using siRNA. We observed that HPAECs lacking *Egfr*, placed either in normoxia or hypoxia, are less angiogenic (Fig S3A) and more apoptotic (Fig S3B) than ECs transfected with scrambled siRNA. These data are consistent with what we observed when we knocked down *AREG* in vitro. In addition, we measured the gene expression of the other EGFr ligands such as *Egf*, *Egf-Hb*, and *Tgfa* in the lungs of normoxic and hypoxic mice (Fig S3C). We did not see any statistical difference in the expression of these ligands in normoxia v. hypoxia. In aggregate, these findings indicate that the Amphiregulin/EGFR signaling controls PAEC survival and proliferation.

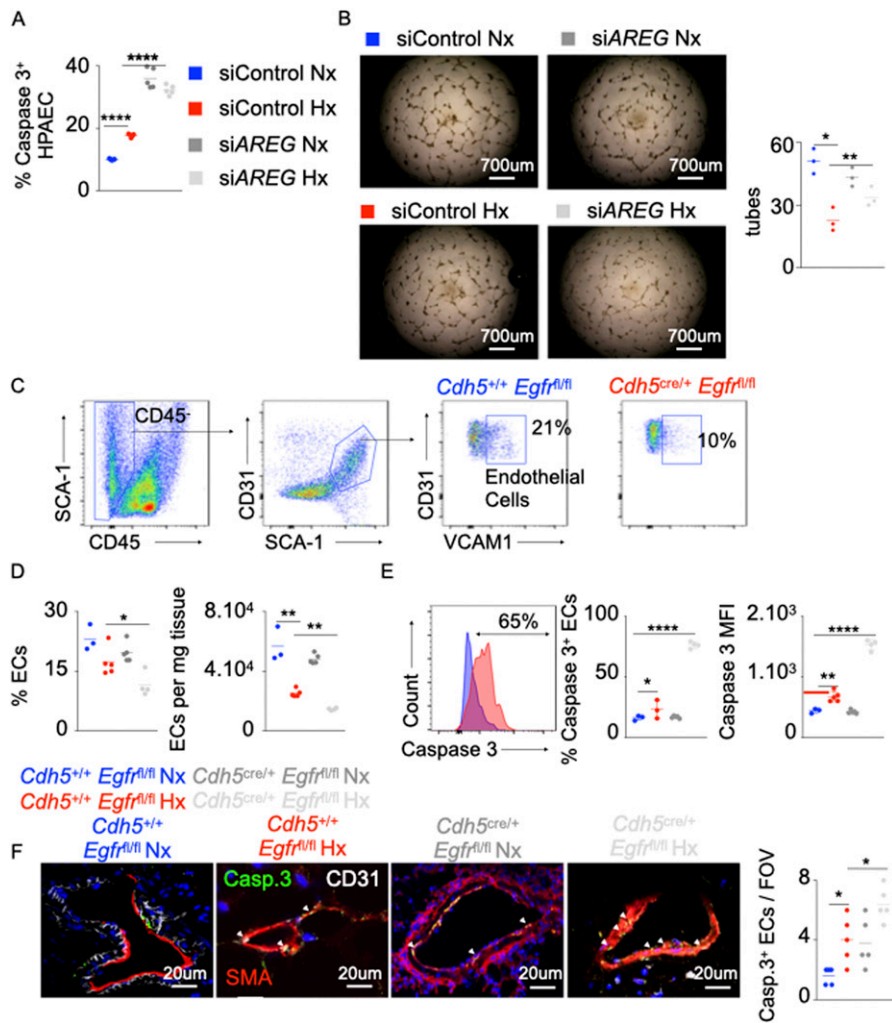

**Figure 2.   The loss of *AREG* and Amphiregulin receptor epidermal growth factor receptor (EGFR) increases EC apoptosis.**
**(A, B)** Pulmonary arterial endothelial cells were transfected with either scrambled siRNA (si*CTL*) or siRNA against *AREG* (si*AREG*) and placed in hypoxia or normoxia. **(A)** The cells were stained for caspase 3, and apoptosis was measured by flow cytometry.
**(B)** Pulmonary arterial endothelial cells were plated on Matrigel, and tubular structures were photographed 4–6 h after plating. The number of tubes was quantified. **(C, E)** *Cdh5*$^{+/+}$ *Egfr*$^{fl/fl}$ and *Cdh5*$^{cre/+}$ *Egfr*$^{fl/fl}$ mice were placed in normoxia or hypoxia for 21 d, and lung vascular endothelial cells were analyzed by flow cytometry. **(C, D)** Flow cytometric quantification of lung vasculature endothelial cells. **(E, F)** The percentage of apoptotic lung vascular endothelial cells was assessed using caspase 3 staining by flow cytometry (E) and confocal microscopy (F). n = 5 per condition. Data are shown as mean. *$P < 0.05$, **$P < 0.01$, ****$P < 0.001$.

## The loss of *AREG* and its receptor *EGFR* in ECs increases their inflammatory phenotype

Various studies reported that resting and healthy endothelial cells prevent blood coagulation, control blood flow, and inhibit inflammation (Pober & Sessa, 2007). Endothelial cell dysfunction characterized by augmented apoptosis is associated with inflammation (Chavakis & Dimmeler, 2002). As discussed above, the disruption of Amphiregulin/EGFR signaling exacerbated PAEC apoptosis in hypoxic conditions. However, if loss of EGFR in PAECs results in exaggerated inflammation in PH is not known. *AREG* silencing in PAECs significantly elevated IL-1β, IL-6, and TNF-α concentrations (Fig S4A) and the expression of the genes encoding these cytokines under hypoxia (Fig 3A). Next, we assessed inflammation in the lungs of *Cdh5*$^{cre/+}$ *Egfr*$^{fl/fl}$ mice that lack *Egfr* in PAECs in normoxia and hypoxia. These mice displayed elevated expression of inflammatory cytokines in the lungs compared with *Cdh5*$^{+/+}$ *Egfr*$^{fl/fl}$ mice (Fig 3B). Given prior paradoxical links of Amphiregulin to fibrosis (Lu et al, 2006; de Vries & Noelle, 2010; Zaiss et al, 2013; Liu et al, 2016b, 2018, 2020), we sought to delineate the contributions of Amphiregulin in lung fibrosis in the context of PH. We first checked the expression of *TGFB, SMAD1, SMAD5,* and *SMAD9,*

which are involved in fibrosis, in normoxic and hypoxic human PAECs transfected with si*AREG*. We found that the expression of these genes was increased in hypoxic PAECs treated with si*AREG* compared with control, highlighting the importance of *AREG* in vitro in modulating fibrosis (Fig S4B). We found similar increase of these genes in the lungs of *Cdh5*$^{cre/+}$ *Egfr*$^{fl/fl}$ mice housed under hypoxic conditions compared with *Cdh5*$^{+/+}$ *Egfr*$^{fl/fl}$ mice (Fig S4C). These data indicate that Amphiregulin reduces the expression of the genes responsible for fibrosis in pulmonary endothelial cells. As a whole, these data indicate that Amphiregulin and EGFR are important for maintaining an anti-inflammatory phenotype of PAECs.

## The loss of *AREG* or its receptor *EGFR* in PAECs recruits inflammatory myeloid cells into the lungs in hypoxic mice

One of the hallmarks of PH is local lung inflammation mediated by the influx of inflammatory myeloid cells, such as monocytes, into the lungs (Amsellem et al, 2017; Florentin et al, 2018). The contribution of Amphiregulin/EGFR in cellular inflammation in PH has not been studied. We observed that siRNA-mediated knockdown of *AREG* in PAECs significantly increased the expression of *CCL1, CCL2, CXCL2,* and *CXCL3* (Fig 4A). These chemokines promote myeloid cell

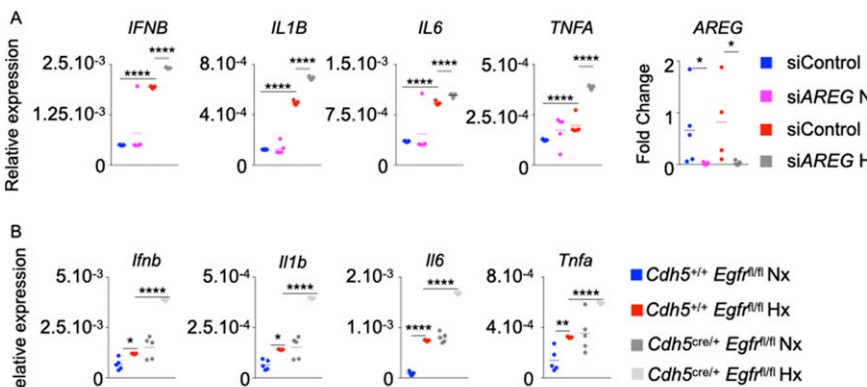

**Figure 3. The loss of *AREG* and Amphiregulin receptor epidermal growth factor receptor (EGFR) in ECs increases their inflammatory phenotype.**
**(A)** Pulmonary arterial endothelial cells were transfected with either scrambled siRNA (siCTL) or siRNA against AREG (si*AREG*) and placed in normoxic or hypoxic conditions. *IFNB*, *IL1B*, *IL6*, *TNFA*, and *ARE* expression was assessed by qRT-PCR. **(B)** *Cdh5*^+/+ *Egfr*^fl/fl and *Cdh5*^cre/+ *Egfr*^fl/fl mice were placed in normoxia or hypoxia for 21 d, and lungs were harvested. *Ifnb*, *Il1b*, *Il6*, and *Tnfa* expression was evaluated in whole lungs by qRT-PCR. **(A, B)** n = 5 samples (A)/mice (B) per condition. Data are shown as mean. *P < 0.05, **P < 0.01, ****P < 0.001.

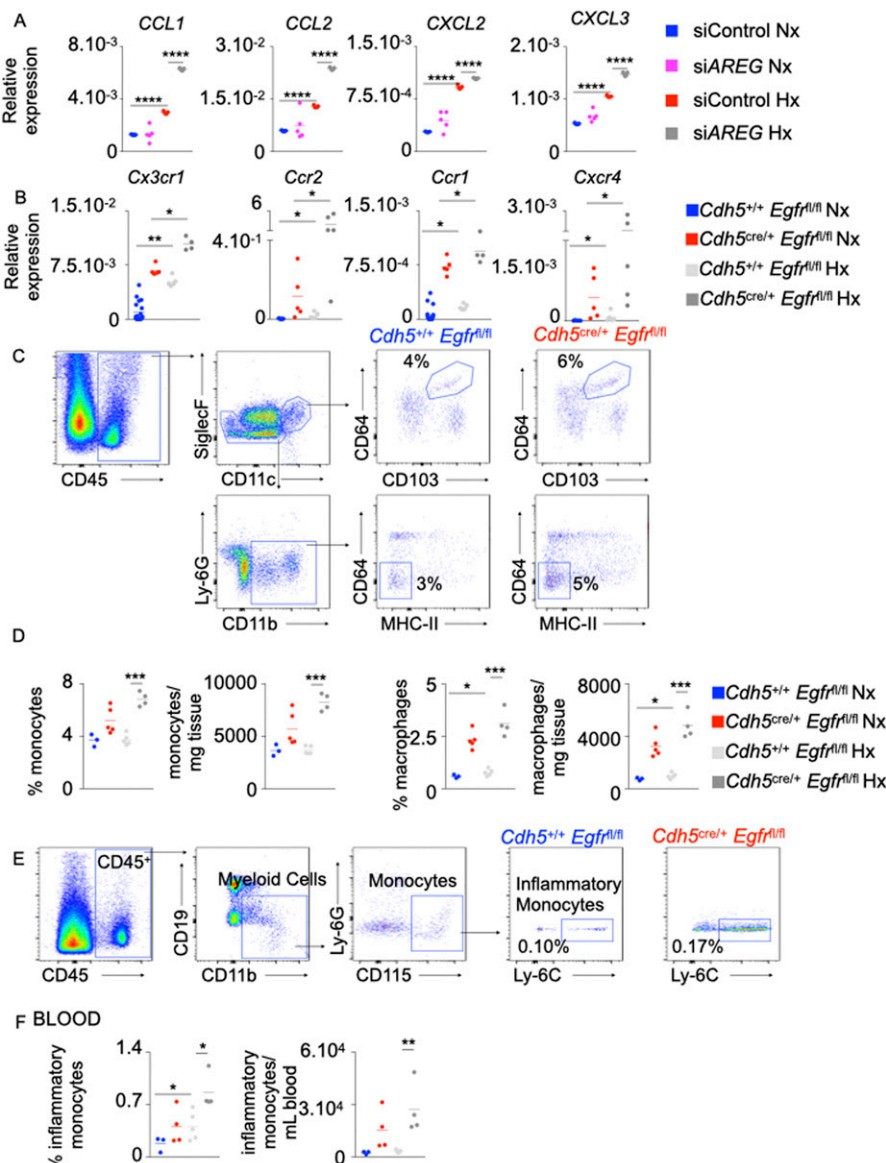

**Figure 4. The loss of *AREG* and *Egfr* in pulmonary arterial endothelial cells recruits inflammatory myeloid cells into the lungs in hypoxic mice.**
**(A)** Human pulmonary arterial endothelial cells were transfected with either scrambled siRNA (siCTL) or siRNA against AREG (si*AREG*) and placed in normoxic or hypoxic conditions. *CCL1*, *CCL2*, *CXCL2*, and *CXCL3* expression was measured by qRT-PCR. **(B)** *Cdh5*^+/+ *Egfr*^fl/fl and *Cdh5*^cre/+ *Egfr*^fl/fl mice were placed in normoxic or hypoxic conditions for 21 d, and lungs were harvested. *Cx3cr1*, *Ccr2*, *Ccr1*, and *Cxcr4* expression was assessed in whole lungs by qRT-PCR. **(C, D, E, F)** Number and frequency of alveolar macrophages (CD45^+, CD11c^+, CD103^+, and CD64^+) and monocytes (CD45^+, Siglec F^−, CD11b^+, MHC-II^−, and CD64^−) were assessed in the lungs (C, D) and blood (E, F) of hypoxic *Cdh5*^+/+ *Egfr*^fl/fl and *Cdh5*^cre/+ *Egfr*^fl/fl mice by flow cytometry. **(A, B, C, D, E, F)** n = 5 replicates (A) and 5 mice (B, C, D, E, F) per condition. Data are shown as mean. *P < 0.05, **P < 0.01, ***P < 0.005, ****P < 0.001.

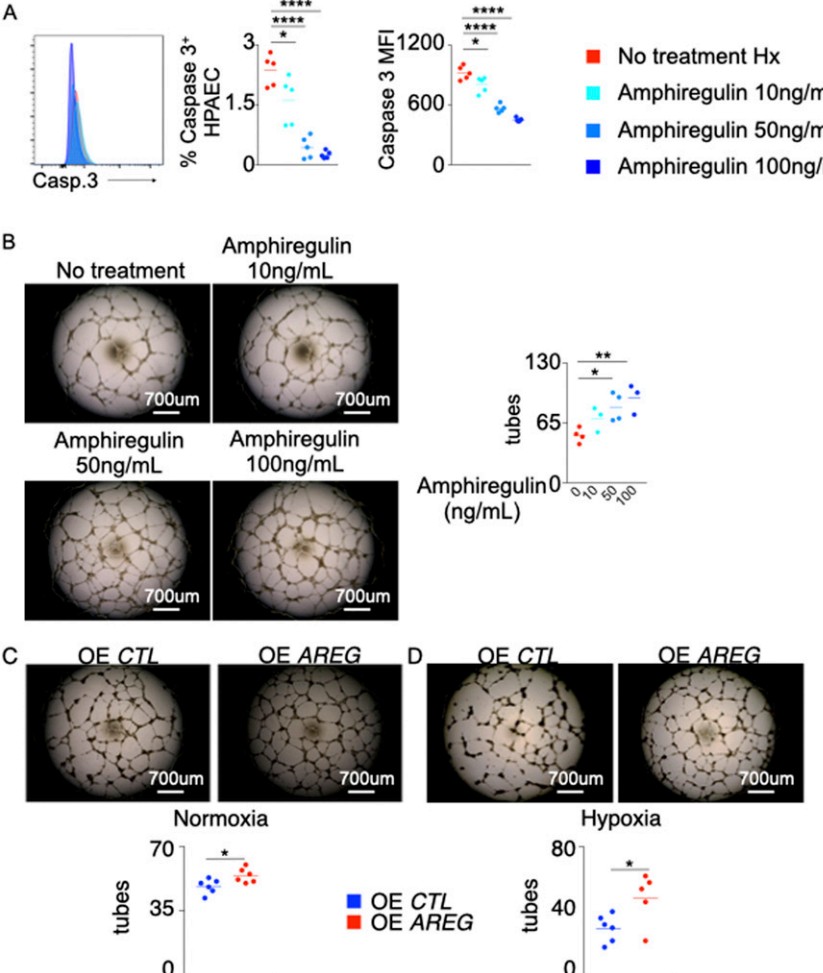

**Figure 5. Amphiregulin treatment decreases pulmonary arterial endothelial cell (PAEC) apoptosis and increases tube formation.**
PAECs were treated with increasing concentrations (10–100 ng/ml) of recombinant Amphiregulin or vehicle and placed under hypoxic conditions. **(A)** PAEC apoptosis was assessed by measuring caspase 3+ cells and caspase 3 mean fluorescence intensity (MFI) by flow cytometry. **(B)** The cells were plated on Matrigel after the treatment, and tube formation was assessed. **(C, D)** PAECs were transfected with either an empty plasmid (OE CTL) or a plasmid encoding AREG (OE AREG) and placed in either normoxic (C) or hypoxic (D) conditions. The tube formation ability of the transfected PAECs was assessed. n = 5 replicates per condition. Data are shown as mean. *P < 0.05, **P < 0.01, ****P < 0.001.

influx at the sites of inflammation. Correspondingly, the lungs of $Cdh5^{cre/+}$ $Egfr^{fl/fl}$ mice showed higher expression of chemokine receptors such as $Cx_3cr1$, $Ccr2$, $Ccr1$, and $Cxcr4$ (Fig 4B). Consistent with high expression of the chemokines in PAECs in the absence of AREG and the chemokine receptors in PAECs of $Cdh5^{cre/+}$ $Egfr^{fl/fl}$ mice, there was a higher abundance of monocytes and macrophages in the lungs of mice lacking Egfr in ECs compared with control (Fig 4C and D). In addition, to understand if Egfr deficiency controls macrophage phenotype, we stained lung sections from $Cdh5^{+/+}$ $Egfr^{fl/fl}$ and $Cdh5^{cre/+}$ $Egfr^{fl/fl}$ mice with antibodies against iNOS and Arg-1, a pro-inflammatory and pro-resolution marker, respectively. We found that lung macrophages of $Cdh5^{cre/+}$ $Egfr^{fl/fl}$ mice expressed lower amounts Arg1 and higher levels of iNOS, indicating that the macrophages in these mice are more inflammatory (Fig S5A and B). Of note, Egfr deficiency in EC also increased circulatory inflammatory monocyte frequency and number (Fig 4E and F). Then, we interrogated the possible role of exogenous Amphiregulin produced by recruited immune cells in the development of PH, which could neutralize or counter any potential deficiency in the endothelial cells. To this end, we first measured the expression of AREG in leukocytes and PAECs under normoxia and hypoxia by qRT-PCR. We observed that leukocytes expressed

AREG at high levels (Fig S5C). To understand if Amphiregulin secreted by leukocytes could counteract the effects of AREG deficiency in endothelial cells, we co-cultured PAECs after AREG silencing with leukocytes isolated from human blood. We found that the presence of leukocytes did not have any impact on apoptosis of normoxic and hypoxic human PAECs (Fig S5D) or their angiogenic ability (Fig S5E). Altogether, these data suggest that the expression of AREG/EGFR in PAECs prevents the recruitment of inflammatory monocytes into the lungs in PH.

### Amphiregulin treatment decreases PAEC apoptosis and increases tube formation

Many studies have reported that increased AREG expression stimulates cell migration and proliferation as well as reduces apoptosis in various diseases including cancers (Fontanini et al, 1998; Jiang et al, 2019). Here, we wanted to discern whether recombinant Amphiregulin treatment in hypoxic PAECs would rescue angiogenesis and prevent apoptosis. To this end, PAECs were treated with various concentrations of recombinant Amphiregulin under hypoxia. Hypoxic PAECs treated with recombinant Amphiregulin were less apoptotic than untreated hypoxic PAECs (Figs 5A and S5F).

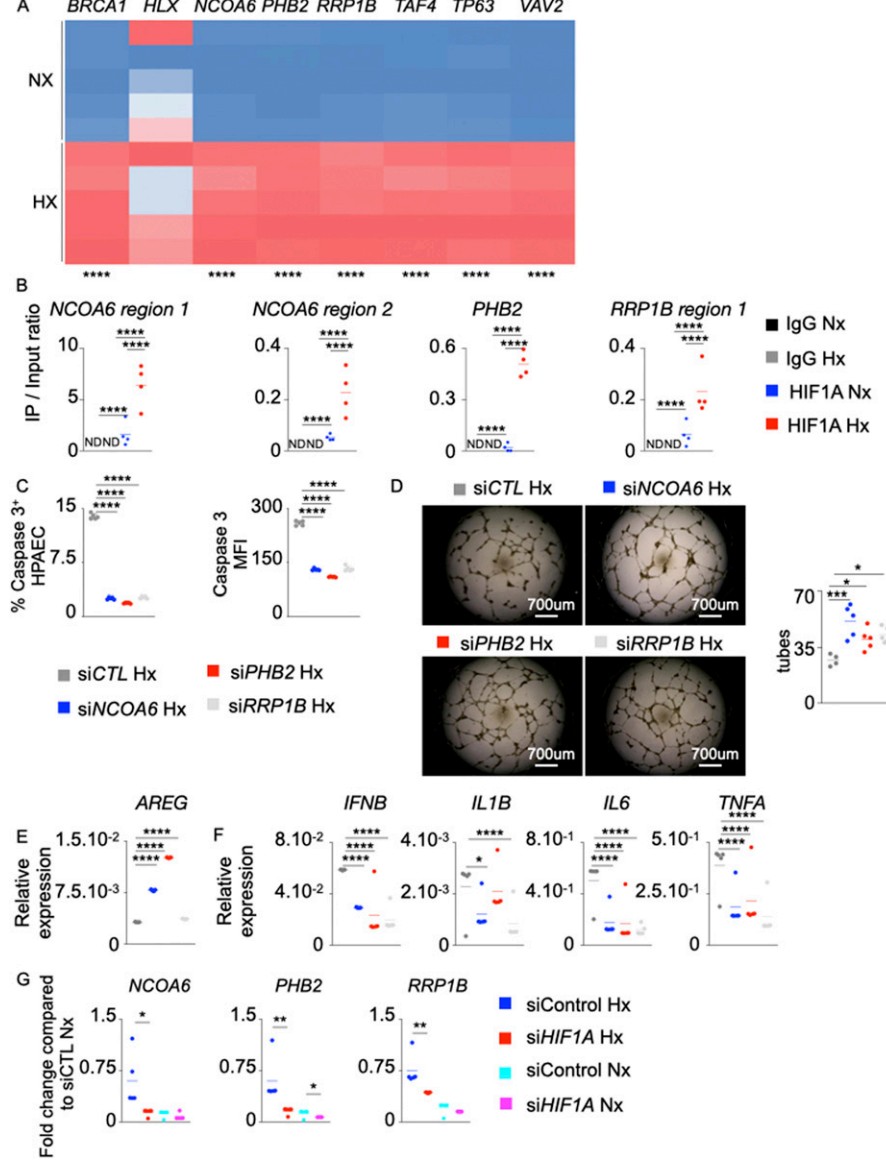

Figure 6. Hypoxia inducible factor-1α (HIF-1α) negatively regulates *AREG* expression in pulmonary arterial endothelial cells (PAECs) under hypoxic conditions.
**(A)** Heat map showing *BRCA1, HLX, NCOA6, PHB2, RRP1B, TAF4, TP63,* and *VAV2* expression assessed by qRT-PCR in normoxic and hypoxic PAECs. **(B)** The binding of HIF-1α to the *NCOA6, PHB2,* and *RRP1B* promoter regions in PAECs was assessed by ChIP qPCR. **(C, D, E, F, G)** PAECs were transfected with scrambled siRNA (si*CTL*) or siRNA against *NCOA6* (si*NCOA6*), *PHB2* (si*PHB2*), or *RRP1B* (si*RRP1B*) and placed in hypoxia for 24 h. **(C)** Apoptotic PAECs were quantified by measuring caspase 3⁺ cells and caspase 3 MFI by flow cytometry. **(D)** PAECs were plated on Matrigel, and tube formation was assessed. **(E, F)** *AREG* (E), *IFNB, IL1B, IL6,* and *TNFA* (F) expression was assessed by qRT-PCR. **(G)** PAECs were transfected with either scrambled siRNA (si*CTL*) or siRNA against *HIF1A* (si*HIF1A*) and placed in hypoxia or normoxia for 24 h. *NCOA6, PHB2,* and *RRP1B* expression was quantified by qRT-PCR. n = 5 replicates per condition. Data are shown as mean. *P < 0.05, **P < 0.01, ***P < 0.005, ****P < 0.001.

Correspondingly, recombinant Amphiregulin exposure increased the tube formation ability of PAECs under hypoxic conditions (Fig 5B). Conversely, we observed that both hypoxic and normoxic HPAECs overexpressing *AREG* showed greater tube formation ability compared with HPAECs transfected with an empty vector (Fig 5C and D).

### Hypoxia Inducible Factor-1 alpha (HIF-1α) negatively regulates *AREG* expression in hypoxic PAECs

To understand the mechanisms of *AREG* down-regulation in endothelial cells under hypoxic conditions, we used IPA to identify possible inhibitors of *AREG*. We found seven possible candidates including BRCA1 (Lamber et al, 2010), HLX (Martin et al, 2013), NCOA6 (Qi et al, 2004), PHB2 (Park et al, 2011), RRP1B (Lee et al, 2014), TAF4 (Fadloun et al, 2007), TP63 (Wang et al, 2011), and VAV2

(Menacho-Márquez et al, 2013) (Fig S6A). All these genes except *HLX* were induced in PAECs under hypoxic conditions (Fig 6A). In silico analysis revealed that the promoter regions of *PHB2, RRP1B,* and *NCOA6* carry binding motifs for HIF-1α (Fig S6B). We experimentally demonstrated the binding of HIF-1α to the promoter regions of *PHB2, RRP1B,* and *NCOA6*in PAECs (Fig 6B). Interestingly, we observed increased HIF-1α binding to the promoter regions of these genes in hypoxic conditions. Silencing these genes in PAECs decreased hypoxia-mediated apoptosis (Figs 6C and S6C), increased tube formation ability (Fig 6D), augmented *AREG* expression (Fig 6E), and decreased PAEC inflammation (Fig 6F). Although there was a modest increase in *AREG* expression in si*RRP1B*-treated PAECs, hypoxia-induced cytokine production was blunted. Of note, knocking down these genes in normoxic human PAECSs also decreased their tube formation ability (Fig S6D). However, the treatment did not have a significant impact on

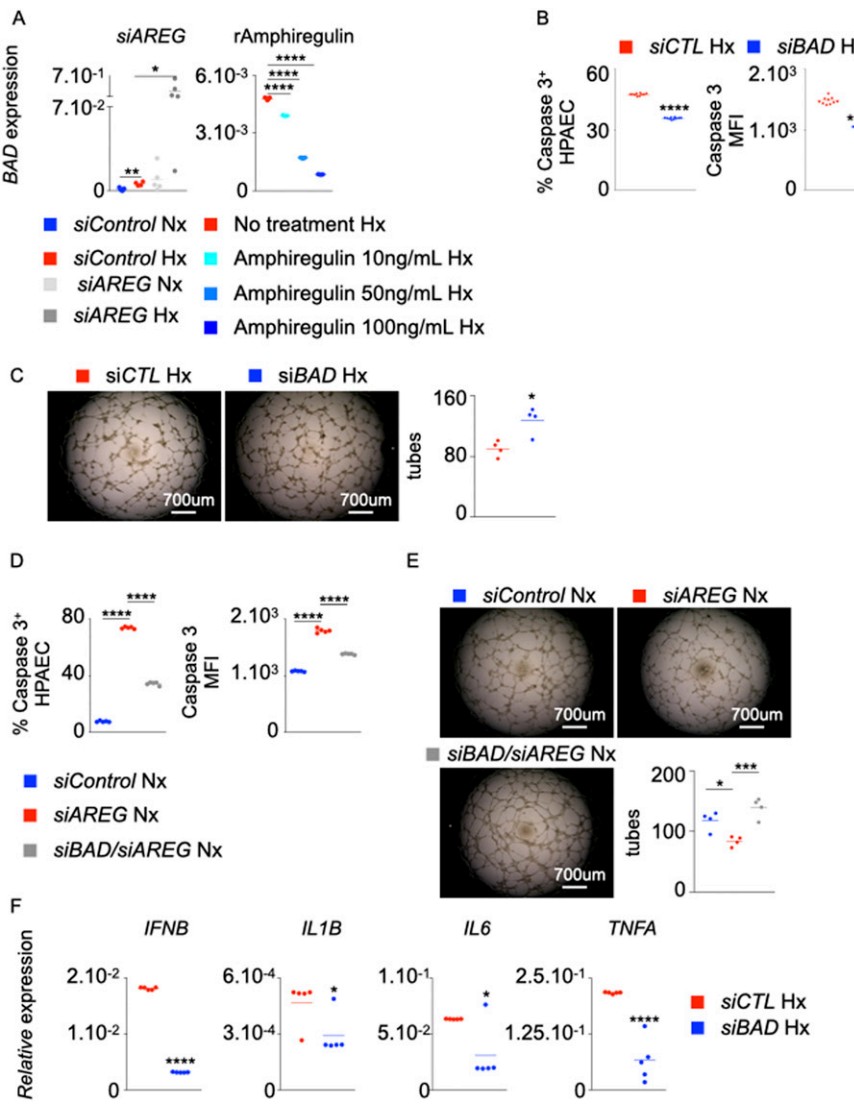

**Figure 7. BAD is essential for exaggerated inflammation, enhanced apoptosis, and suppressed tube formation ability of ECs in the absence of *AREG*.**

**(A)** *BAD* expression in pulmonary arterial endothelial cells (PAECs) was quantified after *AREG* silencing (left panel) and recombinant Amphiregulin treatment (right panel). **(B, C, F)** PAECs were transfected with either scrambled siRNA (si*CTL*) or siRNA against *BAD* (si*BAD*) and placed in hypoxia for 24 h. **(B, C)** PAECs apoptosis was assessed by measuring caspase 3⁺ and caspase 3 MFI cells by flow cytometry (B), and tube formation ability was determined by a Matrigel assay (C). **(D, E)** PAECs were transfected with either scrambled siRNA (si*CTL*), siRNA against *AREG* (si*AREG*) or siRNA against both *AREG* and *BAD* (si*BAD/AREG*) and placed in normoxic conditions for 24 h. **(D, E)** Apoptosis (D) and tube formation (E) were examined. **(F)** *IFNB*, *IL1B*, *IL6*, and *TNFA* expression was assessed by qRT-PCR. n = 5 replicates per condition. Data are shown as mean. *P < 0.05, ***P < 0.005, ****P < 0.001.

*AREG* expression (Fig S6E), indicating HIF-1α–mediated augmentation of the activities of these genes in hypoxia. In addition, siRNA knockdown of *HIF1A* decreased the expression of *NCOA6*, *PHB2*, and *RRP1B* (Fig 6G); however, the treatment increased *AREG* expression (Fig S6F) in PAECs in hypoxic conditions, confirming the regulation of these genes by HIF-1α. To further understand if HIF1α-mediated AREG down-regulation is PHB2-, NCOA6-, and RRP1B-dependent, we silenced these genes in PAECs overexpressing HIF1A. Such knockdown abrogated HIF1α-induced *AREG* down-regulation (Fig S6G). Altogether, these data suggest that HIF-1α suppresses *AREG* expression in hypoxic conditions by activating NCOA6, PHB2, and RRP1B.

### AREG deficiency depends on BAD for exaggerated inflammation, enhanced apoptosis, and depressed angiogenic potential of PAECs

We sought to investigate EGFR/Amphiregulin downstream signaling in PH patients. Our in silico analysis revealed *BAD* as a gene downstream to Amphiregulin/EGFR signaling, controlling cellular apoptosis. si*AREG* treatment augmented the expression of BAD mRNA (Fig 7A, left panel) and protein (Fig S7A) in PAECs cultured under normoxic and hypoxic conditions. Conversely, the exposure of PAECs to recombinant Amphiregulin diminished *BAD* levels in these cells in hypoxic (Fig 7A, right panel) and normoxic conditions (Fig S7B). In addition, siRNA knockdown of *BAD* in hypoxic PAECs reduced hypoxia-induced apoptosis (Fig 7B), improved their tube formation ability in hypoxia (Fig 7C) and decreased the number of recruited inflammatory immune cells (Fig S7C). Moreover, si*BAD* significantly prevented si*AREG*-mediated apoptosis (Fig 7D) and reversed the decrease of normoxic EC tube formation by *AREG* knockdown (Fig 7E). In addition, *BAD* knockdown in PAECs decreased their inflammatory phenotype in hypoxia, as evidenced by diminished mRNA (Fig 7F) and protein (Fig S7D) levels of inflammatory cytokines such as IFN-β, IL-1β, IL-6, and TNF-α. In aggregate, these data demonstrate that *AREG* deficiency depends critically on *BAD* expression for enhanced

PAEC apoptosis, increased inflammation, and suppressed tube formation ability.

## Discussion

Genetic and environmental stimuli trigger PAEC apoptosis (White et al, 2014; Diebold et al, 2015; Vaillancourt et al, 2015), which is one of the inciting factors driving pulmonary vasculature remodeling and PAH. However, the mechanisms of endothelial apoptosis in PAH are understudied. Our analysis of RNA sequencing data comparing PAEC from healthy and PAH patients suggested that Amphiregulin is a potential regulator of endothelial survival. Although Amphiregulin is known to act as a pro-survival molecule, the roles of endothelial *AREG* and its receptor *EGFR* have not been studied. In this study, we sought to elucidate the role of the EGFR ligand Amphiregulin in controlling endothelial pathobiology in PH. In this study, we characterized endothelial apoptosis and inflammation in PH and identified Amphiregulin and EGFR as key molecules regulating endothelial survival and suppressing inflammation in hypoxic conditions and PH. In addition, our study highlighted the orchestration of pulmonary remodeling and hemodynamic manifestations in PH by endothelial EGFR. Mechanistically, we showed that *AREG* expression and inflammatory monocyte recruitment is dependent on HIF1α-mediated NCOA6, PHB2, and RRP1B expression. Last, we showed that the regulation of apoptosis and inflammatory chemokine production by BAD in absence of *AREG* resulting in exacerbation of PH pathogenesis (Fig S7E). In aggregate, these results define a protective role of endothelial Amphiregulin and EGFR in PH, suggesting the notion of Amphiregulin as a therapy of PH for further investigation.

In addition, we acknowledge that PH can be generated through a multitude of mechanisms including chronic hypoxia, thromboembolic disorders, left heart failure, and primary remodeling of the pulmonary vasculature in PPH. We used a hypoxic mouse model for the present study. This mouse model recapitulates the WSPH Group 3 PH. However, this model also is known to recapitulate some aspects of WSPH Group 1 PH (PAH), as it has been widely used to understand inflammatory mechanisms that are important in both groups 1 and 3 PH (Stenmark et al, 2009; Gomez-Arroyo et al, 2012; Ikeda et al, 2019). In addition, we have used two different animal models of PAH: (A) *Il6*[tg] mice expressing *Il6* in pneumocytes that develop severe angioproliferative PAH in hypoxic conditions and (B) Monocrotaline (MCT) exposure in rats exhibiting endothelial dysfunction followed by robust angioproliferative remodeling in small pulmonary arterioles.

The Amphiregulin/EGFR axis has been shown to play a pivotal role in the pathogenesis of various inflammatory diseases (Rayego-Mateos et al, 2018; Stolarczyk & Scholte, 2018). Mechanistically, Amphiregulin binds to the EGFR and activates downstream signaling pathways, such as AKT, MEK1/2, ERK1/2, and NF-κB pathways (Heldin et al, 1998; Heldin & Westermark, 1999; Rosenkranz & Kazlauskas, 1999). Our data show that normoxic mice lacking *Egfr*, specifically in endothelial cells, display greater systemic inflammation because of increased numbers of recruited pro-inflammatory leukocytes such as monocytes and macrophages, and heightened endothelial cell death compared with wild-type

normoxic mice. Interestingly, such increased endothelial inflammation and apoptosis do not result in worse RVSP in absence of hypoxia, indicating that *EGFR* deficiency alone without noxious environmental trigger is not sufficient to drive PH pathogenesis. In the context of hypoxia, the proliferation of cultured human pulmonary microvascular endothelial cells has been shown to depend on EGFR activation (Toby et al, 2010; Pool et al, 2019). In line with this finding, our in vitro studies indicate that Amphiregulin produced by ECs acts in an autocrine fashion and promotes cellular survival and proliferation. We observed that the presence of leukocytes does not reverse the decreased ability of endothelial tube formation in absence of *AREG*. Although amphiregulin secreted by leukocytes can affect endothelial tube formation capability, there are other factors secreted by leukocytes that alters angiogenic ability of ECs. Furthermore, when cultured in contact with HPAECs lacking AREG, inflammatory cells will heighten inflammation in ECs, which may impair angiogenesis (Hanumegowda et al, 2012; Quarck et al, 2015). The protective autocrine function of Amphiregulin is relevant to a key observation made by Rhodes et al (2015), who reported decreased expression in IQSEC1, a guanidine nucleotide exchange protein recruited by EGFR, in PAH patients (Rhodes et al, 2015). Yet, in contrast to prior in vitro findings reporting up-regulation of EGF/EGFR in hypoxic endothelial cells (Toby et al, 2010; Pool et al, 2019), we found that the expression of EGFR was attenuated in pulmonary endothelial cells in human and murine subjects with PH. Importantly, whereas the in vitro response to hypoxia may be more one-sided and may reflect more uncomplicated phenotypes, the evolution of endothelial cells in vivo in PH is known to be complex, including initial apoptosis followed by development of hyperproliferative cells (Michelakis, 2006). Nonetheless, a similarly complex endothelial evolution of EGFR expression and activity may underlie this discrepancy, and future time-course studies of EGFR in vivo should be considered.

EGFR has multiple ligands. Besides Amphiregulin, EGFR interacts with EGF, TGF-α, and heparin-binding EGF (Wieduwilt & Moasser, 2008). Binding of these ligands to EGFR results in different physiological responses (Makki et al, 2013; Singh et al, 2016). Our study does not decipher the contributions of these EGFR ligands in PH. Specifically, Amphiregulin is involved in the pathogenesis of various cardiovascular diseases. Amphiregulin has been shown to increase cardiac fibrosis and aggravate cardiac dysfunction in a mouse model of myocardial infarction (Liu et al, 2018). We have not assessed the role of Amphiregulin produced by PAECs in vivo, and PAEC-specific *AREG*-deficient mice will be required to answer this question. Moreover, future studies are warranted to decipher the contributions of Amphiregulin produced by other pulmonary cells such as smooth muscle cells (Heialy et al, 2013; Deacon & Knox, 2015), which are crucial to the development of PH.

It has also been reported that TGF-α–dependent EGFR signaling promotes PAH (Le Cras et al, 2003) and that the inhibition of the EGFR attenuated PH in a rat model (Merklinger et al, 2005). Our findings that endothelial-specific deficiency of EGFR in fact promotes PH point to a cell type-specific role for this pleiotropic receptor. Correspondingly, in the development of hepatocellular carcinoma, EGFR has been shown to elicit opposing physiological functions depending on cell type (Timchenko, 2015). Future work in animal models carrying genetic depletions of EGFR across different

cellular compartments in the lung may reveal further distinct actions of EGFR, converging upon the final manifestation of PH. In addition, it has been proposed that at the initial stage of PH, pulmonary ECs undergo apoptosis followed by the development of hyperproliferative cells (Michelakis, 2006). During the initial stage of PH, low endothelial expression of amphiregulin/EGFR contributes to elevated EC apoptosis, whereas in the latter stage, amphiregulin promotes EC proliferation and fibrosis. Moreover, EGFR exerts opposing effects depending on cell type as stated above (Timchenko, 2015). Thus, our findings support the notion that targeting *AREG* expressed by endothelial cells in the early stage of PH in patients could be beneficial. This could be achieved by using siRNA against *AREG* formulated in 7C1 nanoparticles, which specifically target endothelial cells (Dahlman et al, 2014; Sager et al, 2016).

Unexpectedly, in endothelial cells, we found that EGFR is involved in suppressing inflammation. Genetic and siRNA-mediated endothelial deletion of *EGFR* and *AREG* resulted in increased production of inflammatory cytokines and chemokines responsible for myeloid cell recruitment. In agreement with this, the lungs of $cdh5^{cre/+}$ $Egfr^{fl/fl}$ mice harbored augmented number of monocytes compared with age-matched control. Conversely, in an animal model of atherosclerosis, Egfr expressed in atheromas facilitates monocyte recruitment and their subsequent differentiation into macrophages (Lamb et al, 2004). The discrepancy in Egfr-mediated inflammatory potential can again be explained by tissue and cell specificity. Studies from our and other groups have shown that monocytes infiltrated into the lungs in PH increase inflammation (Rabinovitch et al, 2014; Pugliese et al, 2015; Amsellem et al, 2017; Kumar et al, 2017; Florentin et al, 2018) and worsen PH burden. Similarly, aggravation of PH in the absence of EGFR in PAECs could result from not only accelerated apoptosis but also heightened inflammation. However, mice lacking monocytes and *Egfr* will be necessary to precisely dissect these mechanisms further. In addition, we used $Cdh5^{cre/+}$ $Egfr^{fl/fl}$ mice, which have a conditional deficiency of *Egfr* in endothelial cells. We acknowledge that we did not use inducible Cre mice, such as Cdh5 ER-Cre, which would allow us to avoid any developmental adaptation. Further analysis using the latter mouse model is required to decipher the possibility of developmental adaptation that a conditional deletion of *Egfr* in endothelial cells might generate.

While investigating the mechanisms of *AREG* down-regulation in hypoxic conditions, we observed that the expression of various upstream transcription co-regulators, including *NCOA6*, *RRP1B*, and *PHB2* was augmented in hypoxic PAECs. These molecules are found to be up-regulated in the hypoxic microenvironments in tumors (Cheng et al, 2014; Xu et al, 2019; Wu et al, 2020a). Interestingly, the functions of these transcription factors in inflammation and PH are largely unknown. We observed that HIF-1α increases the expression of these transcription factors by directly binding to their promoter regions. Furthermore, silencing of *NCOA6*, *RRP1B*, and *PHB2* abrogated *AREG* down-regulation, endothelial apoptosis, and inflammation in hypoxic condition. Additional studies will be necessary to ascertain the exact roles of these transcription factors in PH pathogenesis.

Finally, with the help of IPA and in vitro assays, we found that BCL2-Associated Agonist of Cell Death (BAD), a molecule downstream to EGFR/Amphiregulin, plays a pivotal role in enhancing apoptosis and

inflammation and suppressing the angiogenic potential of ECs in the absence of *AREG*/*EGFR*. Correspondingly, BAD expression has been shown to be up-regulated in various subtypes of PH (Liu et al, 2016c; Deng et al, 2017; Opitz & Kirschner, 2019). Specifically, increased *Bad* expression has been associated with heightened endothelial apoptosis and lung remodeling in rat models of chronic thromboembolic pulmonary hypertension (CTEPH) (Deng et al, 2017) and World Symposium on Pulmonary Hypertension (WSPH) Group 3 PH (Liu et al, 2016c). Thus, our report connects these two seemingly disparate pathways of pathogenic molecules, describing an *AREG*-mediated BAD down-regulation and prevention of endothelial apoptosis in PH. Further studies using endothelial-specific deletion of *Bad* will be required to evaluate BAD as a potential druggable target to dampen inflammation, promote endothelial survival, and reduce PH pathogenesis.

# Materials and Methods

### Animals

All animal experiments were conducted following NIH guidelines under protocols approved by the Institutional Animal Care and Use Committee of the University of Pittsburgh. Adult females C57BL/6 wild-type and lung-specific Il-6–overexpressing transgenic mice ($Il6^{tg}$) (10–12 wk old) were obtained from the Jackson Laboratory and maintained under a standard light cycle (12 h light/dark). $Il6^{tg}$ mice express pulmonary-specific *Il-6*, leading to secondary systemic increases of Il-6 (Steiner et al, 2009). $Egfr^{fl/fl}$ mice were gifts from Dr. Matthias Nahrendorf. $cdh5^{cre/+}$ male and $Egfr^{fl/fl}$ female mice were bred to generate $Cdh5^{+/+}$ $Egfr^{fl/fl}$ and $Cdh5^{cre/+}$ $Egfr^{fl/fl}$ mice. To induce PH, mice were placed in hypoxic chambers under 10% $O_2$ for 3 wk. Adult, 8–10-wk-old male Sprague–Dawley rats were obtained from Charles River laboratory and maintained under a standard light cycle (12 h light/dark). To induce PH, rats were injected with monocrotaline (60 mg/kg).

### Organ harvesting and flow cytometry

Mice were euthanized and perfused thoroughly with 30 ml of ice-cold PBS through the left ventricle. One lobe of the lungs was harvested, minced and digested in an enzymatic mixture of collagenase I, collagenase XI, DNase I, and hyaluronidase (Sigma-Aldrich) for leukocyte analysis, and collagenase IV and DNase I for endothelial cell separation, under agitation at 37°C for an hour. Cells were passed through 40-$\mu$m cell strainers, washed in 10 ml of FACS buffer that is made of PBS containing 0.5% bovine serum albumin and centrifuged at 4°C. Peripheral blood was collected through cardiac puncture, and RBC lysis buffer was used to lyse erythrocytes for 3 min at room temperature. After lysis, FACS buffer was added and the cells were centrifuged. Supernatant was discarded and pellet was dissolved in FACS buffer. Total viable cell numbers per mg of tissue for lungs and per mL of blood were counted using a hemocytometer after dissolving the single cell suspension in Trypan Blue (Cellgro, Mediatech Inc.). The staining of the single cell suspension was performed with the following

antibodies diluted in FACS buffer. These antibodies were purchased from eBioscience, BioLegend, or BD Biosciences. The following panel of antibodies was used to analyze myeloid cell population: anti-CD45.2 (104), CD11b (M1/70), CD115 (AFS98), Ly6G (1A8), and Ly6C (AL-21). The following panel of antibodies was used to analyze endothelial cell population: anti-CD45.2 (104), Sca-1 (M1/70), CD31 (MEC13.3), and Vcam1 (429). Monocytes were identified as CD11b$^+$, Ly6G$^-$, and CD115$^+$. Endothelial cells were defined as CD45.2$^-$, Sca-1$^+$, CD31$^+$, and Vcam1$^+$. Apoptosis experiments were carried out using PE Active Caspase 3 Apoptosis kit from BD Biosciences. A Fortessa Flow Cytometer (BD) was used to acquire data. Data were analyzed with FlowJo software (Tree Star).

## Immunofluorescence

After 30 ml of cold PBS perfusion through the left ventricle, one lobe of the lungs was excised and fixed in 4% (PFA, methanol-free, Electron Microscopy Science) for 1 h and then incubated in 30% sucrose PBS at 4°C. Then, the tissue was embedded in Tissue-Tek OCT compound (Sakura Finetek) and frozen on dry ice. For immunofluorescence staining, 0.1% Triton X-100 PBS was used to permeabilize 20-$\mu$m-thick sections for 1 h at room temperature. To prevent off-site binding of antibodies, the sections were blocked in 2% BSA PBS for 1 h. Next, sections were incubated with anti-SMA-Cy3 (Millipore) for 1 h to detect smooth muscle cells in lung vasculature. The sections were subsequently washed with PBS containing 0.5% BSA, and counterstained and fixed with vector shield DAPI to visualize nuclei, and images were taken using Confocal laser scanning immunofluorescence microscopy (CLSM). Image analysis was performed using ImageJ software (Fiji). The degree of pulmonary arteriolar muscularization was assessed in lung sections stained for $\alpha$-SMA by the calculation of the proportion of fully and partially muscularized peripheral (<100 mm in diameter) pulmonary arterioles among total peripheral pulmonary arterioles, as previously described (Ingersoll et al, 2010). Medial thickness was also measured in $\alpha$-SMA–stained vessels (<100 mm in diameter) using ImageJ software (Fiji) and expressed as arbitrary units. All measurements were performed blinded to condition.

## qRT-PCR

One half of a lobe of the lungs was flash frozen in liquid nitrogen. RNeasy RNA isolation kit (QIAGEN) was used to extract total mRNA. Total mRNA concentration was measured using a NanoDrop device. Then, cDNA was generated from 100 ng of mRNA with the high capacity RNA to cDNA synthesis kit (Applied Biosystems). SYBRgreen primers (IDT) were used to quantify the expression of genes using Quantitative RT-PCR. Data were expressed as Δ Ct values of genes normalized to the house-keeping gene $\beta$-actin. Final results were then showed as relative expression of genes compared with $\beta$-actin. The relative expression of genes was calculated as follows: Relative expression$_{gene}$ = POWER(2 – (Ct$_{gene}$ – Ct$_{beta\ actin}$)).

## Catheterization and hemodynamics

The right jugular vein was dissected. Salivary and lymphatic tissues were separated out to visualize and isolate a section of the vessel.

Using 4-0 silk suture, the cranial end of the vessel was tied, and a loose tie on the caudal end of the vessel was made. An incision large enough to pass and insert the catheter between the two ligatures was made using a 22-G needle. The catheter was advanced towards the heart until desired and stable signal appears. RVSP was recorded. Heart was flushed with 10 ml of PBS, and the right ventricle (RV) was separated from the left ventricle (LV). Both ventricles were weighed, and RV/LV ratios were calculated.

## RNAseq data and ingenuity pathway analysis

Dr. Marlene Rabinovitch kindly provided the expression values of genes differentially expressed in PAEC of patients with PAH and age-matched control patients. CLC Genomics and IPA (QIAGEN Inc.) (Krämer et al, 2013) were used to analyze the data. We assessed the fold change in the expression of the genes involved in endothelial cell survival. Genes with at least twofold differences were selected for the pathway analysis. Using IPA, we focused on gene sets whose expression was down-regulated in PH patients.

## ChIP sequencing

To assess the interaction between HIF1-$\alpha$ and NCOA6, PHB2, and RRP1B, we first located the promoter regions of NCOA6, PHB2, and RRP1B with the help of UCSC genome browser Website (http://genome.ucsc.edu, University of California Santa Cruz). Then, we looked for potential HIF-1α–binding sites onto these promoter regions using Transfac software (GeneXplain). The primers for each HIF1-$\alpha$ binding site were designed using the NCBI Primer Blasts. PAECs were plated onto 10 cm dishes and for 24 h in hypoxia or normoxia. Cells were then harvested and resuspended in PBS at a concentration of 1 × 10$^6$ cells/ml. Cells were then fixed and DNA was extracted and immunoprecipitated by a ChIP grade HIF1-$\alpha$ antibody (Rb polyclonal; Novusbio) as previously described (Dahl & Collas, 2008). Finally, real-time PCR was run to quantify the amount of HIF1A bound to NCOA6, PHB2, and RRP1B promoter regions in each condition.

## PAEC culture and AREG inhibition

Human PAECs were obtained from Promocell and grew in Endothelial Basal Medium-2 (Promocell). The cells were further expanded to reach passage 3. At passage 3, cells were transfected with siRNA against *AREG* (IDT) for 72 h. After 48 h of siRNA treatment, cells were put in hypoxic chamber for 24 h. Cells were then analyzed for gene expression, apoptosis and in vitro tube formation assay.

## Amphiregulin treatment

PAECs were cultured for 96 h in the presence or absence of 10, 50, or 100 ng/ml of recombinant Amphiregulin (Sigma-Aldrich) that has been previously reconstituted in filtered PBS containing 0.1% BSA. Cells were then used for apoptosis and in vitro angiogenesis assays.

### In vitro angiogenesis assays

Matrigel with reduced growth factors was pipetted into pre-chilled 96-well plate (50 $\mu$l Matrigel per well) and polymerized for 30 min at 37°C. PAECs following different treatments (2 × 10$^4$ cells per well) were resuspended in 100 $\mu$l of basic media, and seeded in Matrigel-coated 96-well plate. After 4–6 h of incubation, tubular structures were photographed using Olympus inverted fluorescent microscope with a 20× magnification. The number of branch points was quantified in technical triplicate.

### Statistical analysis

Data were compiled in PRISM software (GraphPad), and statistics were generated and represented as mean ± SEM. Normality of data distribution was determined by Shapiro–Wilk testing. For normally distributed data, statistical significance between two categories of analyzed samples was calculated using two-tailed $t$ test. For non-normally distributed data, statistical significance was calculated using Mann–Whitney U test. For multiple category comparisons, one-way ANOVA was used with post hoc Bonferroni testing. Differences with $P$-values less than 0.05 were considered statistically significant.

### Study approval

All experimental procedures involving the use of human lung tissue included the relevant receipt of written informed consent and were approved by the Committee for Oversight of Research and Clinical Training Involving Decedents No. 101, at the University of Pittsburgh, as well as the Institutional Review Board of the University of Pittsburgh No. REN17020169/IRB020810. All experimental procedures involving the use of human peripheral blood included the relevant receipt of written informed consent and were approved by the Institutional Review Board of the University of Pittsburgh No. REN16070123/PRO11070366 as well as the Institutional Review Board of the University of Pittsburgh No. REN17030011/IRB0306040. Ethical approval for this study and informed consent conformed to the standards of the Declaration of Helsinki.

## Data Availability

For data sharing and information, please contact Dr. Partha Dutta, the corresponding author at duttapa@pitt.edu.

## Supplementary Information

## Acknowledgements

This work was supported by National Institute of Health grants R00HL121076-03 (to P Dutta), R01HL143967 (to P Dutta), R01HL142629 (to P Dutta), and R01AG069399 (to P Dutta), AHA Transformational Project Award (19TPA34910142 to P Dutta), AHA Innovative Project Award (19IPLOI34760566 to P Dutta), and ALA Innovation Project Award (IA-629694 to P Dutta) as well as the VMI Postdoctoral Training Program in Translational Research and Entrepreneurship in Pulmonary and Vascular Biology T32 funded by the National, Heart, Lung and Blood Institute (NHLBI) (to J Florentin). The work was also supported by NIH grants R01 HL124021 and HL 122596, as well as the American Heart Association Established Investigator Award 18EIA33900027 (SY Chan). We thank the NIH supported microscopy resources in the Center for Biologic Imaging (NIH grant 1S10OD019973-01). We thank Dr. Marlene Rabinovitch, Stanford University, who kindly provided RNA sequencing data. Additionally, we acknowledge Dr. Matthias Nahrendorf, Massachusetts General Hospital for Systems Biology, Harvard University, for kindly providing us with the $Egfr^{fl/fl}$ mice. We thank the Center for Organ Recovery & Education (CORE) as well as organ donors and their families for the generous donation of tissues used in this study. We used Biorender application (https://biorender.com) to generate Figs 1A and S7E.

### Author Contributions

J Florentin: conceptualization, data curation, formal analysis, investigation, and writing—original draft, review, and editing.
J Zhao: data curation.
Y-Y Tai: data curation.
W Sun: data curation.
LL Ohayon: data curation.
SP O'Neil: data curation.
A Arunkumar: data curation.
X Zhang: data curation.
J Zhu: methodology.
Y Al Aaraj: resources.
A Watson: resources.
J Sembrat: resources.
M Rojas: resources.
SY Chan: resources, funding acquisition, and writing—review and editing.
P Dutta: conceptualization, resources, supervision, funding acquisition, project administration, and writing—review and editing.

### Conflict of Interest Statement

SY Chan has served as a consultant for United Therapeutics and Acceleron Pharma. SY Chan is a director, officer, and shareholder in Synhale Therapeutics. SY Chan has held research grants from Actelion, Bayer, and Pfizer. SY Chan has filed patent applications regarding the targeting of metabolism in pulmonary hypertension. The authors declare no other conflicts of interest.

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
