## [Reviewer comments · Life Science Alliance]

Life Science Alliance

Amphiregulin loss in endothelial cells drives inflammation and apoptosis in pulmonary hypertension

Jonathan Florentin, Jingsi Zhao, Yi-Yin Tai, Wei Sun, Lee Ohayon, Scott O'Neil, Anagha Arunkumar, Xinyi Zhang, Janhui Zhu, Yassmin Al Aaraj, Annie Watson, John Sembrat, Mauricio Rojas, Stephen Chan, and Partha Dutta

DOI: <https://doi.org/10.26508/lsa.202101264>

Corresponding author(s): Partha Dutta, University of Pittsburgh Medical Center

Review Timeline:

Submission Date:	2021-10-19
Editorial Decision:	2021-11-22
Revision Received:	2022-05-16
Editorial Decision:	2022-06-06
Revision Received:	2022-06-09
Accepted:	2022-06-09

Transaction Report:

November 22, 2021

Re: Life Science Alliance manuscript #LSA-2021-01264-T

Dr. Partha Dutta
University of Pittsburgh Medical Center
200 Lotrop Street
BST1720.1
Pittsburgh 15213

Dear Dr. Dutta,

Thank you for submitting your manuscript entitled "Amphiregulin loss in endothelial cells drives inflammation and apoptosis in pulmonary hypertension" to Life Science Alliance. The manuscript was assessed by expert reviewers, whose comments are appended to this letter. We invite you to submit a revised manuscript addressing the Reviewer comments.

Thank you for this interesting contribution to Life Science Alliance. We are looking forward to receiving your revised manuscript.

Sincerely,

B. MANUSCRIPT ORGANIZATION AND FORMATTING:

Reviewer #1 (Comments to the Authors (Required)):

Florentin et al sought to examine the role of amphiregulin (AREG) in the endothelial cell (EC) dysfunction observed in pulmonary hypertension (PH). The authors show that AREG is downregulated using previously published RNAseq data from pulmonary artery ECs (PAECs) from patients with PH, and that humans and mice with PH show downregulation of its receptor epidermal growth factor receptor (EGFR). In vitro studies implicated hypoxia inducible factor-1 α (HIF-1 α) induced upregulation of NCOA6, PHB2, and RRP1B in AREG downregulation and subsequent augmentation of BCL2-associated agonist of cell death (BAD). Collectively, this incited EC apoptosis, diminished angiogenesis, and increased chemokine production and myeloid cell recruitment. Mice with EC deletion of *Egfr* exposed to hypoxia were shown to have increased RVSP and pulmonary remodeling. The authors propose a protective role for endothelial AREG and EGFR in PH, and a potential therapeutic role for amphiregulin.

A considerable amount of work has been performed by the authors, demonstrating new aspects of regulation, and downstream and functional effects of amphiregulin signaling in PAECs. This is a strength of the manuscript. Nonetheless, the role of amphiregulin in the pathophysiology of PH in vivo is not conclusive on the basis of these studies. While the isolated cell studies focus on AREG, the in vivo studies focus on *Egfr*. Although AREG is a ligand for *Egfr*, there are also several other ligands that can activate this tyrosine kinase receptor; hence, it is unclear whether EC-specific *Egfr* knockout mice can be used to model AREG downregulation in vivo. There is no direct evidence presented that AREG is downregulated in lungs of the PH models used. Moreover, the EC *Egfr* knockout mice appear to have a baseline pulmonary phenotype with increase pulmonary remodeling, EC apoptosis, and inflammation (Figures 1G, 2F, 4B, D, F) - this may confound the phenotype related to hypoxia. The assessment of PH in the other two rodent models (IL6 Tg mouse hypoxia and rat monocrotaline injection) are underdeveloped. Thus, the conclusions related to PH pathophysiology are somewhat premature.

The authors should also address the following:

1. What is the level of amphiregulin gene and protein in the lung in the PH models and human PH, and is there confirmation of reduced endothelial protein expression?
2. Please show data as mean +/- SD as well as individual data points in the figures. Please clarify statistical methodology - in several experiments (e.g. Figure 1F-G), two-way ANOVA seems more appropriate.
3. Figure 1D - please clarify this is histological staining in lungs from patients with PH. The figure text states normoxia and hypoxia - do you mean control and PH lungs? Please provide details in the methods as to how these human specimens were obtained and subject characteristics, etiology of PH, etc.
4. Figure S1D-E - EGFR and Ki67 staining is difficult to appreciate in the examples provided. Please provide better quality immunostaining. There are no details provided of the IL6 Tg mice or the monocrotaline rat model in the methods.
5. Figures 1F-G - the authors should provide representative hemodynamic traces and RV gravimetric data. Also, provide details as to how the lung remodeling scores were calculated and the parameters used.
6. Figure 2 - to support the in vivo data in EC-specific *Egfr* knockout mice, can the authors provide data on the effects of siEGFR (normoxia and hypoxia) on the Matrigel tube formation assay? In figure 2F, it is hard to appreciate caspase and CD31 co-staining in the images provided.
7. Figure S3 - panel D why does leukocyte AREG have no effect on EC apoptosis whereas recombinant AREG does? Also, please show data from hypoxic conditions in this panel. Panel E - although stated otherwise in the text, the presence of leukocytes appears to significantly decrease tube formation under hypoxic conditions and siAREG treatment.
8. Figure 4 - please provide more detail as to rationale for gating strategies for myeloid cells in panel C.
9. Figure 4D - lung macrophages and monocytes appear augmented in EC specific EGFR KO mice under baseline normoxic conditions suggesting a baseline inflammatory phenotype. What are the implications of this finding? Please discuss.
10. Figure S5 - Panel A: BAD protein expression under hypoxic conditions is not shown, although it is stated in the text. Panel C: the experimental design for immune cell recruitment is not clear.
11. Figure 7 - labeling of bars in in panel A is incomplete.
12. Discussion, page 15. The authors posit cell-type specific opposing beneficial and detrimental responses of EGFR signaling in PH. What are the implications of such dichotomous responses for amphiregulin as a potential therapeutic in PH?
13. Methods - please specify the sex and age of the mice and rats used in the PH study.

Reviewer #2 (Comments to the Authors (Required)):

In this manuscript, Florentin et al investigated the effects of decreases in Amphiregulin in endothelial cells on inflammation, apoptosis and vascular formation in pulmonary hypertension and demonstrated that Amphiregulin and EGFR may be potential therapeutic targets to reduce inflammation in PH. The paper is interesting and well written, however there are several critical points to be addressed.

1. The tube formation assay is conducted to evaluate the vascular formation in the study, but the pictures are poor and it is hard to see the difference among the conditions. To convince the effects on angiogenesis, better images should be provided. How is the tube formation analyzed? tube numbers?
2. AREG levels are examined only at the mRNA levels. Protein levels in ECs and lung tissues would be helpful to show the involvement of AREG in ECs. Are there any other reports showing that AREG is expressing in ECs?
3. The effects of AREG overexpression in ECs should be analyzed.
4. The IF images in Figs.1, 2, S1, S3 are poor and not convincing. Better images would be necessary.

Minor:

1. The patient numbers in Fig. 1A do not match the legend and text.

Reviewer #1 (Comments to the Authors (Required)):

Florentin et al sought to examine the role of amphiregulin (AREG) in the endothelial cell (EC) dysfunction observed in pulmonary hypertension (PH). The authors show that AREG is downregulated using previously published RNAseq data from pulmonary artery ECs (PAECs) from patients with PH, and that humans and mice with PH show downregulation of its receptor epidermal growth factor receptor (EGFR). In vitro studies implicated hypoxia inducible factor-1 α (HIF-1 α) induced upregulation of NCOA6, PHB2, and RRP1B in AREG downregulation and subsequent augmentation of BCL2-associated agonist of cell death (BAD). Collectively, this incited EC apoptosis, diminished angiogenesis, and increased chemokine production and myeloid cell recruitment. Mice with EC deletion of *Egfr* exposed to hypoxia were shown to have increased RVSP and pulmonary remodeling. The authors propose a protective role for endothelial AREG and EGFR in PH, and a potential therapeutic role for amphiregulin.

A considerable amount of work has been performed by the authors, demonstrating new aspects of regulation, and downstream and functional effects of amphiregulin signaling in PAECs. This is a strength of the manuscript. Nonetheless, the role of amphiregulin in the pathophysiology of PH in vivo is not conclusive on the basis of these studies. While the isolated cell studies focus on AREG, the in vivo studies focus on *Egfr*. Although AREG is a ligand for *Egfr*, there are also several other ligands that can activate this tyrosine kinase receptor; hence, it is unclear whether EC-specific *Egfr* knockout mice can be used to model AREG downregulation in vivo.

We thank the reviewer for his comment. We agree

that, since there are multiple ligands of *Egfr*, it is difficult to delineate the role of Areg by using endothelial cell-specific *Egfr* KO mice. In this regard, endothelial cell-specific Areg KO mice will be ideal. However, these mice are not commercially available. Since we observed *Egfr* down regulation in pulmonary endothelial cells in vivo in PAH patients and different animal models of PH (Figure 1D and E in the manuscript), and endothelial *Egfr* is protective against PH, we decided to understand the impact of this receptor on endothelial biology *in vitro*. We silenced *Egfr* in HPAECs in vitro using siRNA and observed that HPAECs lacking *Egfr*, placed either in normoxia or hypoxia, are less angiogenic (Rebuttal Figure 1A, Figure S3A in the manuscript, line #157) and more apoptotic (Rebuttal Figure 1B, Figure S3B in the manuscript, line #158) than ECs transfected with scrambled siRNA. These data are consistent with what we observed when we knocked down *Areg* in vitro.

Additionally, we measured the gene expression of the other EGFr ligands such as *Egf*, *Egf-Hb* and *Tgfa* in the lungs of normoxic and hypoxic mice. We did not observe any statistical difference in the expression of these Egfr ligands in normoxia v. hypoxia (Rebuttal Figure 1C, Figure S3C in the manuscript, line #161). Additionally, we have acknowledged the limitations of our study in respect to understanding the role of Areg *in vivo* in Discussion.

There is no direct evidence presented that AREG is downregulated in lungs of the PH models used.

We thank the reviewer for pointing this out. We now show the data that AREG expression in pulmonary ECs of hypoxic mice and PAH patients is downregulated compared to the controls (Rebuttal Figures 2A and 2B, Supplemental Figure S1B and S1C in the manuscript, line 116). These data are in line with our observations of reduced AREG gene expression *in vitro* in HPAECs and in patients with PH (Figures 1B and 1C in the manuscript).

Moreover, the EC Egfr knockout mice appear to have a baseline pulmonary phenotype with increase pulmonary remodeling, EC apoptosis, and inflammation (Figures 1G, 2F, 4B, D, F) - this may confound the phenotype related to hypoxia.

As this reviewer correctly pointed out, our data show that normoxic mice lacking Egfr specifically in endothelial cells have increased systemic inflammation, due to increased numbers of recruited proinflammatory leukocytes such as monocytes and macrophages, and heightened endothelial cell death compared to wildtype normoxic mice. Interestingly, these increased endothelial inflammation and apoptosis do not result in worse RVSP

in absence of hypoxia, indicating that EGFR deficiency alone without noxious environmental trigger is not sufficient to drive PH pathogenesis. We have now included this explanation in Discussion (line 310 – 316 in the manuscript).

The assessment of PH in the other two rodent models (IL6 Tg mouse hypoxia and rat monocrotaline injection) are underdeveloped. Thus, the conclusions related to PH pathophysiology are somewhat premature.

To specifically address this concern, we have quantified the levels of Egfr, caspase 3 and Bad in all rodent models we have used in the present study: hypoxic mice, *Il6^{tg}* mice, and monocrotaline-injected rats. We showed a concomitant decrease in Egfr expression and increase in caspase 3 levels in all the rodent PH models. Moreover, there were inverse correlations between Egfr and caspase 3 expressing ECs (Rebuttal

Figure 3A, Figure S2A in the manuscript, line#123). Additionally, we found an increased Bad expression in pulmonary ECs across all rodent PH models, suggesting that Egfr regulates the expression of Bad, protecting ECs against apoptosis. (Rebuttal Figure 3B, Figure S2B in the manuscript, line#126).

The authors should also address the following:

1. What is the level of amphiregulin gene and protein in the lung in the PH models and human PH, and is there confirmation of reduced endothelial protein expression?

We found significant diminution of AREG gene expression in mice and patients with PH (Figure 1B and 1C in the manuscript). Our new data now show that AREG MFI and the percentage of AREG⁺ pulmonary endothelial cells follow the same trend (Rebuttal Figures 2A & 2B, Supplemental Figure S1B and S1C in the manuscript, line 116).

2. Please show data as mean +/- SD as well as individual data points in the figures. Please clarify statistical/methodology - in several experiments (e.g. Figure 1F-G), two-way ANOVA seems more appropriate.

Figures throughout the manuscript have been modified and statistical analysis have been clarified as per the Reviewer's comments (line 519-525 in the manuscript).

3. Figure 1D - please clarify this is histological staining in lungs from patients with PH. The figure text states normoxia and hypoxia - do you mean control and PH lungs? Please provide details in the methods as to how these human specimens were obtained and subject characteristics, etiology of PH, etc.

We apologize for this mistake. We have now changed the labels to "Control" and "PH". The origin of the lungs and details regarding human specimens have been added to the methods section as the Reviewer requested.

4. Figure S1D-E - EGFR and Ki67 staining is difficult to appreciate in the examples provided. Please provide better quality immunostaining. There are no details provided of the IL6 Tg mice or the monocrotaline rat model in the methods.

Better quality immunostaining has been added per the reviewer's request (Figures S1B-E). Additionally, we have now included details of the IL6^{tg} mice and monocrotaline treatment (line 412, 414, 415, 418-420 in the manuscript).

5. Figures 1F-G - the authors should provide representative hemodynamic traces and RV gravimetric data. Also, provide details as to how the lung remodeling scores were calculated and the parameters used.

As this Reviewer requested, representative hemodynamic traces have been provided (Rebuttal Figure 4, Figure S2C in the manuscript, line#137). We have also included the methods for calculating the lung remodeling.

Rebuttal Figure 4: Representative hemodynamics traces of hypoxic $Cdh5^{Cre/+} Egfr^{fl/fl}$ and $Cdh5^{+/+} Egfr^{fl/fl}$ mice

6. Figure 2 - to support the in vivo data in EC-specific Egfr knockout mice, can the authors provide data on the effects of siEGFR (normoxia and hypoxia) on the Matrigel tube formation assay? In figure 2F, it is hard to appreciate caspase and CD31 co-staining in the images provided.

We thank the reviewer for his comment. We transfected HPAECs with siEGFR and exposed them to either normoxia or hypoxia for 24 hours. We then performed an *in vitro* tube formation assay. This experiment revealed that the lack of EGFR in HPAECs decreased their angiogenic abilities in both normoxic and hypoxic conditions (Rebuttal Figures 1A, Figure S3A in the manuscript, line#157). Additionally, we have now included better quality images for caspase 3 and CD31 co-staining in Fig. 2F.

7. Figure S3 - panel D why does leukocyte AREG have no effect on EC apoptosis whereas recombinant AREG does? Panel E - although stated otherwise in the text, the presence of leukocytes appears to significantly decrease tube formation under hypoxic conditions and siAREG treatment.

The Reviewer has brought up an interesting point. We observed that the presence of leukocytes does not reverse the decreased ability of endothelial tube formation in absence of *AREG*. Although amphiregulin secreted by leukocytes can affect endothelial tube formation capability, there are other factors secreted by leukocytes that alters angiogenic ability of ECs. Furthermore, when cultured in contact with HPAECs lacking *AREG*, inflammatory cells will heighten inflammation in ECs, which may impair angiogenesis (1, 2). We have now included this explanation in Discussion (lines 320-325 in the manuscript).

Also, please show data from hypoxic conditions in this panel.

The data from hypoxic conditions have been added to Figure S5 (Rebuttal Figure 5, Supplemental Figure 5D lower panel in the manuscript, line#213).

8. Figure 4 - please provide more detail as to rationale for gating strategies for myeloid cells in panel C.

A rationale for the myeloid cell gating strategy for Figure 4C has been added in the figure legend (line 956-959 in the manuscript).

9. Figure 4D - lung macrophages and monocytes appear augmented in EC specific EGFR KO mice under baseline normoxic conditions suggesting a baseline inflammatory phenotype. What are the implications of this finding? Please discuss.

As discussed above, EC-specific Egr KO mice have slight increased inflammation and endothelial apoptosis. However, these mice do not have increased RVSP when they are not exposed to hypoxic condition. These implicates that although EC-specific Egr KO mice are prone to inflammation, hypoxic trigger is required for exaggerated inflammatory phenotype and PH pathogenesis. We have included this in Discussion as the Reviewer suggested (line 310 – 316 in the manuscript).

10. Figure S5 - Panel A: BAD protein expression under hypoxic conditions is not shown, although it is stated in the text. Panel C: the experimental design for immune cell recruitment is not clear.

We apologize for this omission. We have now assessed BAD expression at the protein level by flow cytometry (Rebuttal Figure 6, Supplemental Figure 7A in the manuscript, line#261). Additionally, we have clarified the experimental design for immune cell recruitment in Figure S7C.

11. Figure 7 - labeling of bars in in panel A is incomplete.

The Y axes in Figure 7A have now been labeled.

12. Discussion, page 15. The authors posit cell-type specific opposing beneficial and detrimental responses of EGFR signaling in PH. What are the implications of such dichotomous responses for amphiregulin as a potential therapeutic in PH?

The work described in this manuscript indicates that amphiregulin produced by ECs acts in an autocrine fashion and promotes cellular survival and proliferation. Additionally, it is known that, at initial stage of PH, pulmonary ECs undergo apoptosis followed by the development of hyperproliferative cells (3). During the initial stage of PH, low endothelial expression of amphiregulin/EGFR contributes to elevated EC apoptosis while in the latter stage, amphiregulin promotes EC proliferation and fibrosis. Additionally, EGFR exerts

opposing effects depending on cell type (4). Thus, targeting AREG expressed by endothelial cells in the early stage of PH in patients will be beneficial. This can be achieved by using siRNA against AREG formulated in 7C1 nanoparticles, which, we have shown, specifically target endothelial cells (5) (lines 357-365 in the manuscript).

13. Methods - please specify the sex and age of the mice and rats used in the PH study.

The gender and age of mice and rats used in our study have been specified in the methods section per the Reviewer's request (line 412, 414, 415, 418-420 in the manuscript).

Reviewer #2 (Comments to the Authors (Required)):

In this manuscript, Florentin et al investigated the effects of decreases in Amphiregulin in endothelial cells on inflammation, apoptosis and vascular formation in pulmonary hypertension and demonstrated that Amphiregulin and EGFR may be potential therapeutic targets to reduce inflammation in PH. The paper is interesting and well written, however there are several critical points to be addressed.

1. The tube formation assay is conducted to evaluate the vascular formation in the study, but the pictures are poor and it is hard to see the difference among the conditions. To convince the effects on angiogenesis, better images should be provided. How is the tube formation analyzed? tube numbers?

We have now provided with TIFF images. Additionally, we have added a detailed method for tube formation and number calculations (line #511-516 in the manuscript).

2. AREG levels are examined only at the mRNA levels. Protein levels in ECs and lung tissues would be helpful to show the involvement of AREG in ECs. Are there any other reports showing that AREG is expressing in ECs? **As this Reviewer suggested, we have evaluated AREG protein expression in pulmonary ECs of hypoxic mice and PH patients. By confocal imaging, we observed that AREG expression in both mice and patients with PH is downregulated compared to the controls (Rebuttal Figures 2A and 2B, Figure S1B and S1C in the manuscript respectively, line#116). These data are consistent with the observations of AREG gene expression in mice housed in hypoxia and PH patients.**

3. The effects of AREG overexpression in ECs should be analyzed.

We have now transfected HPAECs with a vector encoding AREG and placed the cells in normoxia or hypoxia for 24 hours. We observed that both hypoxic and normoxic HPAECs overexpressing AREG have a better tube formation ability compared to HPAECs with an empty vector (Rebuttal Figure 7, Figures 5C and D in the manuscript, line#227). Additionally, we have shown that AREG silencing significantly decreased the tube formation ability of HPAECs (Figures 2B and S5E in the manuscript, line#213). These data demonstrate that AREG expression in endothelial cells is crucial to their tube formation ability.

4. The IF images in Figs.1, 2, S1, S3 are poor and not convincing. Better images would be necessary.

Better images have been added per the reviewer's comments.

Minor:

1. The patient numbers in Fig. 1A do not match the legend and text.

The discrepancy has been resolved.

References

1. Quarck R, Wynants M, Verbeken E, Meyns B, Delcroix M. Contribution of inflammation and impaired angiogenesis to the pathobiology of chronic thromboembolic pulmonary hypertension. *European Respiratory Journal*. 2015;46(2):431-43.
2. Hanumegowda C, Farkas L, Kolb M. Angiogenesis in Pulmonary Fibrosis: Too Much or Not Enough? *Chest*. 2012;142(1):200-7.
3. Michelakis ED. Spatio-temporal diversity of apoptosis within the vascular wall in pulmonary arterial hypertension: heterogeneous BMP signaling may have therapeutic implications. *Circulation research*. 2006;98(2):172-5.
4. Timchenko NA. Cell-type specific functions of epidermal growth factor receptor are involved in development of hepatocellular carcinoma. *Hepatology*. 2015;62(1):314-6.
5. Sager HB, Dutta P, Dahlman JE, Hulsmans M, Courties G, Sun Y, et al. RNAi targeting multiple cell adhesion molecules reduces immune cell recruitment and vascular inflammation after myocardial infarction. *Sci Transl Med*. 2016;8(342):342ra80.

June 6, 2022

RE: Life Science Alliance Manuscript #LSA-2021-01264-TR

Dr. Partha Dutta
University of Pittsburgh Medical Center
200 Lotrop Street
BST1720.1
Pittsburgh 15213

Dear Dr. Dutta,

Thank you for submitting your revised manuscript entitled "Amphiregulin loss in endothelial cells drives inflammation and apoptosis in pulmonary hypertension". We would be happy to publish your paper in Life Science Alliance pending final revisions necessary to meet our formatting guidelines.

- please provide your main manuscript text file in editable doc format
- please upload your supplementary figures as single files and add your supplementary figure legends to the main manuscript text, directly under the main figure legends
- please add ORCID ID for corresponding author-you should have received instructions on how to do so
- please add a summary blurb/alternate abstract to our system
- please add the Twitter handle of your host institute/organization as well as your own or/and one of the authors in our system
- please use the [10 author names, et al.] format in your references (i.e. limit the author names to the first 10)
- please double-check your figure callouts in your main manuscript text; you have a callout for Figure S8, but this is neither in the legends nor in the uploaded figures
- please delete the link, <https://www.qiagenbioinformatics.com/products/ingenuity-pathway-analysis>, since company links tend to go stale
- please add a Data Availability Statement to indicate accession information for the RNA-seq and CHIP-seq data

Figure Check:

- please make sure all sets of microscopy images contain a scale bar

A. FINAL FILES:

B. MANUSCRIPT ORGANIZATION AND FORMATTING:

Sincerely,

Reviewer #1 (Comments to the Authors (Required)):

In this revised paper, the authors have responded adequately to all of my original concerns. I have no further comments

Reviewer #2 (Comments to the Authors (Required)):

Authors addressed all my previous concerns.

June 9, 2022

RE: Life Science Alliance Manuscript #LSA-2021-01264-TRR

Dr. Partha Dutta
University of Pittsburgh Medical Center
200 Lotrop Street
BST1720.1
Pittsburgh 15213

Dear Dr. Dutta,

Thank you for submitting your Research Article entitled "Amphiregulin loss in endothelial cells drives inflammation and apoptosis in pulmonary hypertension". It is a pleasure to let you know that your manuscript is now accepted for publication in Life Science Alliance. Congratulations on this interesting work.

DISTRIBUTION OF MATERIALS:

Again, congratulations on a very nice paper. I hope you found the review process to be constructive and are pleased with how the manuscript was handled editorially. We look forward to future exciting submissions from your lab.

Sincerely,
